# RPV Sealing Reliability Estimating Using a New Inconsistent Knowledge Fused Bayesian Network and Weighted Loss Function

**Hao Huang** [1,†], **Ying Luo** [1,2,†], **Caiming Liu** [1], **Yuanyuan Dong** [2], **Xiaoran Wei** [1], **Zhe Zhang** [1], **Xu Chen** [1] and **Kai Song** [1,*]

[1] School of Chemical Engineering and Technology, Tianjin University, Tianjin 300072, China; huanghoward@tju.edu.cn (H.H.); luo_ying@tju.edu.cn (Y.L.); 1018207004@tju.edu.cn (C.L.); wxr_by@tju.edu.cn (X.W.); zhe.zhang@tju.edu.cn (Z.Z.); xchen@tju.edu.cn (X.C.)
[2] Nuclear Power Institute of China, Chengdu 600032, China; npichd10@npic.ac.cn
* Correspondence: ksong@tju.edu.cn
† These authors contributed equally to this work.

**Abstract:** The sealing system performance of a nuclear reactor pressure vessel (RPV) under different working circumstances is vital to the safe operation of the whole nuclear process; however, the mechanism, and especially the relationship among highly related variables for it, has remained unclear. Therefore, a new inconsistent knowledge fused Bayesian network and weighted loss function (iBWL) method was proposed to identify key variables and estimate the reliability of an RPV sealing system. In this method, a new inconsistent knowledge fusion method was proposed to make good use of available priceless field knowledge by considering its reliability and inconsistency seriously. The key variables identified by the Bayesian network structure were then used by a new weighted loss function to estimate the reliability of the RPV sealing system by comprehensively quantifying the deviations of them from their corresponding expected values. It is not only the quantified reliability of RPV sealing that can provide solid information for its operation status but also the weighted loss function can provide clues for how to tune the corresponding parameters to make sure RPV operating has an acceptable status. The application performed on the simulation samples based on the RPV of Liaoning Hongyanhe Nuclear Power Plant and another two RPV units in service strongly proved the outstanding performance of this advanced iBWL method.

**Keywords:** Bayesian network; loss function; nuclear reactor pressure vessel; reliability estimation; sealing system performance





## 1. Introduction

A reactor pressure vessel (RPV) is a part of the pressure boundary of a nuclear reactor coolant system. It is used to support and accommodate the reactor core and prevent the leakage of radioactive substances. A RPV is mainly composed of a closure–head, a cylinder, and a sealing system between them. Compared with the high strength of the shell, the sealing system consisting of a flange connection is the most vulnerable part of the RPV [1]. The fluctuation of the high working pressure and temperature during the service of a RPV, especially during the start–up and shutdown processes, may cause sealing failure, which may lead to the leakage of high–pressure radioactive fluid and other serious accidents [2].

Because a RPV is the only main equipment that cannot be replaced in the whole service life of the nuclear reactor [3]—which highlights the importance of its sealing system—tremendous efforts have been made to improve its sealing performance. Currently these studies are mainly based on finite element analysis (FEA) and can be summarized into two categories: (1) the study of structure parameters under various transient conditions [4] and (2) the development of the sealing ring and its compression rebound characteristics [5].

Given the structure parameters (including the number and positions of bolts, the type of sealing rings, and so on) and working temperature and pressure, FEA is a very powerful tool that can simulate the operation status of a RPV with a high enough accuracy [6]. Since there are not any online measurement methods available, FEA is the major method to calculate bolt preload, the axial separation of the gasket, and other status variables. Although these studies have laid a solid foundation and accumulated priceless knowledge in the mechanism of the RPV sealing system, there is still significant room for improvements, for example, (1) most of these studies have been explored using the FEA method, which means they are univariate analysis. It has not been possible yet to estimate the sealing performance if one or several parameters deviate from their expected or designed values; (2) FEA methods can only calculate the structure parameters (i.e., the axial separation of the flange and so on) of a RPV rather than quantify their influence on sealing performance, not to mention to quantify their comparative importance; and (3) there has been no standard quantitative score to evaluate the reliability of a RPV sealing system under certain circumstances. Using an FEA simulation model as the main method, it is not possible to realize online monitoring, predicted health management, or any other advanced safety management. It is of great need to provide a way to estimate the deviations of both the status variables and structure variables comprehensively not only for reliability estimation during the service time of a RPV but also for the realization of the intrinsic safety design.

Unlike an unmanned aerial vehicle, pump, turbine or other kinds of disposable parts or equipment, the cost, time, and danger are too much to afford for an RPV failure experiment. Consequently, it is a typical unsupervised problem to analyze the relationships among variables. Such kinds of problems exclude ANN (artificial neural networks), CNN (convolutional neural networks), SVM (support vector machine), and other black–box or supervised machine learning methods [7]. It also makes Bayesian network (BN) analysis an optimal option because it is a DAG (directed acyclic graph) based network and can describe the relationships among variables very clearly [8]. Most importantly, BN is one of the few machine learning methods that can integrate expert knowledge to improve accuracy and speed up the learning procedure [9,10]. Additionally, this expert knowledge can help verify the causal relationship recovered by the BN. It has been widely applied in the fields of fault diagnosis and environmental models, reliability estimation, risk assessment, key feature identification to assist mechanism research, and so on [11,12]. Sun et al. reported a new PC–PSO algorithm that integrates human knowledge to learn a BN structure [13]. Eunice et al. incorporated the background knowledge in the form of ancestral constraints into the BN structuring algorithm [14].

However, these works have not considered the uncertainty of knowledge but rather, regarded the expert knowledge as a hard constraint. They assume that the knowledge given by experts can make the causal relationship between nodes in the BN structure clear and reliable, but actually, most knowledge learned from practice is uncertain. Even for the same knowledge, in the same domain, there may exist contradicting qualitative statements on dependency, causality, and parameters over a set of entities [15]. The limitation of the current research is that mechanism analysis cannot effectively guide safety management. There are tremendous mechanism analyses on RPV sealing systems, but few of them are reflected in the practical safety management work. To make good use of this fuzzy and inconsistent domain knowledge and to fill in the gaps between mechanism analysis, safety management and a thorough intrinsic safety design, a new iBWL method was proposed in our study.

There are three contributions of the iBWL method: (1) It emphasizes the uncertainty in the knowledge provided by several experienced experts and faces the challenge of quantifying it. Then it fuses the inconsistent knowledge and analyzes the RPV sealing system using both objective data and subjective knowledge. (2) It identifies key variables by inferring a Bayesian network providing useful guidance in safety management. (3) It estimates the reliability of the sealing system of the RPV caused by the deviation of variables, and this reliability analysis was verified with two sets of real nuclear plant data.

The rest of the paper is structured as follows: Section 2, a brief introduction of the background and related existing methods. Section 3, a description of the proposed iBWL method in detail. Section 4, the results and discussion on the sealing performance of RPV obtained by the proposed method and Section 5, the conclusion.

## 2. Preliminaries and Background

### 2.1. Introduction of the RPV and Its Sealing System

As we mentioned above, RPVs must endure a high internal temperature and strong radioactivity during their long–term service. Compared with the strength and sealing performance of the cylinder, the sealing system (mainly composed of a closure–head, a cylinder and a sealing system arranged between them) at the flange connection is relatively weak. It is easy to cause flange sealing failure with high–pressure fluid leakage under certain fluctuations of working pressure and temperature.

The reliability of the sealing system is related to the system stiffness distribution, deformation, sealing ring performance, surface finish, slot size, transient conditions, heat transfer and other factors [16]. To obtain the deformation and stress distribution of a RPV, researchers generally use FEA to simulate the structure and working status [5]. Because it has not been possible to measure axial separation or other important variables online yet, continuous efforts have been made with FEA to simulate and calculate values of these variables with a high enough accuracy if the internal pressure and temperature have been given.

To analyze the relationships among both the structure and status variables, 699 simulation examples based on the RPV of the Liaoning Hongyanhe Nuclear Power Plant with different dimension parameters were performed using ANSYS 15.0. The operation process of the RPV was simulated by the sequential coupling method. Solid70 was used as a thermal analysis element and Solid185 was used as the structural element for parametric modeling and meshing, respectively. To ensure accuracy and calculation efficiency, the model structure was simplified appropriately [17].

The positions of the variables are shown in Figure 1 and all their definitions are described in Table 1. The structure variables and status variables were all taken into consideration. The axial separation of the flange, the axial separation of the gasket, the flange angle, and other units' displacements in critical positions were chosen as the status variables.

**Table 1.** Description of variables.

| Variables [1] | Description [2] | Variables [1] | Description [2] |
|---|---|---|---|
| FL_ZKL | $\Delta UY_1 - \Delta UY_2$. Axial separation of the flange. | SR2 | The inner diameter of the closure–head. |
| ZX_IN | $\Delta UY_7 - \Delta UY_8$. Axial separation of the inner seal ring. | D17 | The outer diameter of upper cladding. |
| ZX_OUT | $\Delta UY_5 - \Delta UY_6$. Axial separation of the outer seal ring. | D15 | The starting point of the ramp. |
| JX_IN | $\Delta UX_7 - \Delta UX_8$. Radial separation of the inner seal ring. | D14 | The outer diameter of the inner seal groove. |
| JX_OUT | $\Delta UX_5 - \Delta UX_6$. Radial separation of the outer seal ring. | D13 | The pitch diameter of the inner seal ring. |
| ZJ_U | $\tan^{-1}\left(\frac{\Delta UY_3 - \Delta UY_1}{\|\Delta UX_3 + UX_3 - UX_1 - \Delta UX_1\|}\right)$. Upper flange angle.[2] | D12 | The outer diameter of the outer seal groove. |
| ZJ_D | $\tan^{-1}\left(\frac{\Delta UY_4 - \Delta UY_2}{\|\Delta UX_4 + UX_4 - UX_2 - \Delta UX_2\|}\right)$. Lower flange angle.[2] | D11 | The pitch diameter of the outer seal ring. |
| LS_YJL | Bolt preload. | D10 | The outer diameter of the flange of the cylinder. |
| θ | Ramp angle. | D8 | The inner diameter of the cylinder flange. |
| T2 | Wall thickness of the cylinder. | D7 | The inner diameter of cylinder flange. |
| T1 | Wall thickness of closure–head. | D3 | The outer diameter of flange of closure–head. |
| H3 | Height of flange of the cylinder. | D2 | Bolt centerline diameter. |
| H1 | The downward offset of the center of the upper head. | D1 | The inner diameter of flange of closure–head. |

[1] Red color: status variables; blue color: structure variables. [2] UY, UX, $\Delta UY$ and $\Delta UX$ are shown in Figure 1d.

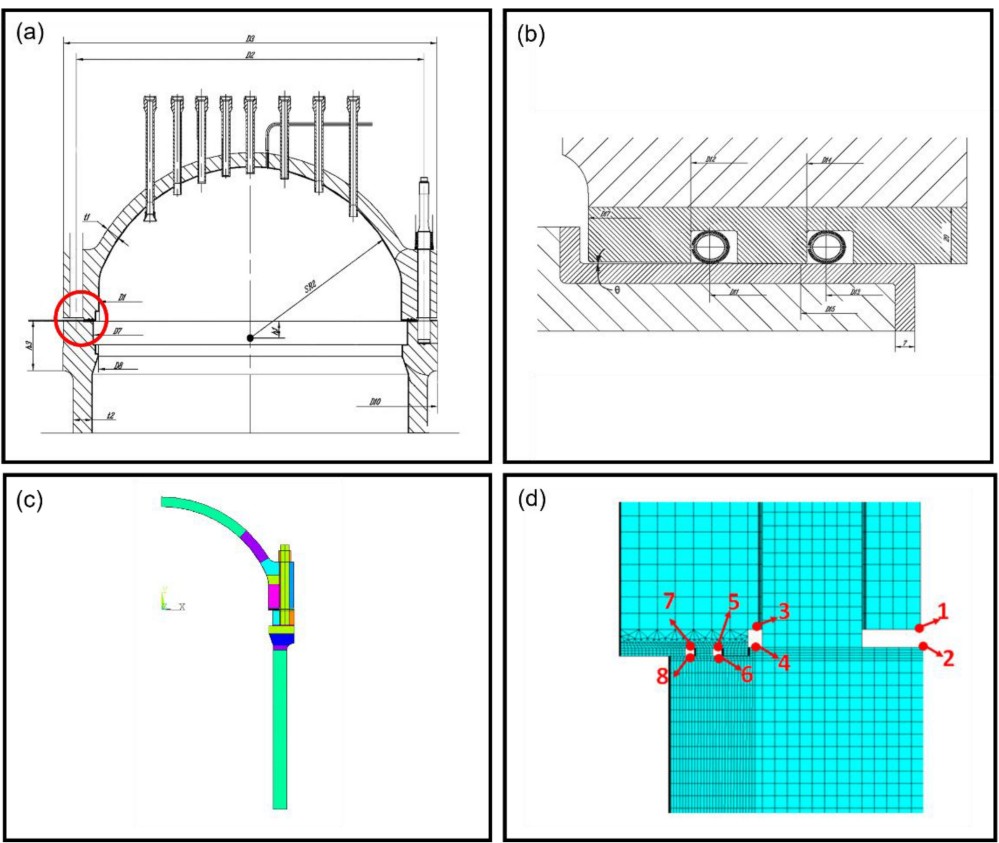

**Figure 1.** (**a**) RPV dimension parameters; (**b**) partially zoomed figure of dimension parameters in sealing position; (**c**) RPV's FEA model; (**d**) special location illustration, $UY_i$, $UX_i$ are the *i*-th unit's Cartesian coordinate positions. $\Delta UY_i$ and $\Delta UX_i$ are the *i*-th unit's displacement in Y and X direction. ($i$ = 1 . . . 8).

### 2.2. The Introduction of Regular BN Algorithms

A Bayesian network (BN) is a probability graph model based on Bayes theory, which can be expressed as $BN = (G, P)$, where $G = (N, E)$ represents a direct acyclic graph (DAG), $P$ is the joint probability distribution, $N = \{x_1, x_2, \ldots, x_n\}$ is the node–set and $E$ is the edge set. The $G$ structure represents the intensity of causality between the nodes/variables $x_1, x_2, \ldots, x_n$ ($n$ is the number of nodes/variables). A directed edge set $E$ indicates the dependency relationship between nodes/variables. In a DAG, if there is a directed edge $x_i \rightarrow x_j$, then $x_i$ is identified as a parent of $x_j$ and $x_j$ is a child of $x_i$. This means $x_j$ is directly affected by $x_i$. $Pa(x_i)$ is used to represent the parent set of $x_i$. The dependence of a node $x_i$ on its parent set $Pa(x_i)$ can be quantified by the conditional probability, expressed as $P(x_i \mid Pa(x_i))$, and the joint probability distribution of all variables in a BN is as [8],

$$P(x_1, x_3, \ldots, x_n) = \prod_{i=1}^{n} P(x_i | Pa(x_i)) \tag{1}$$

For a given dataset $D$, constructing a BN is the task of finding the most suitable network that can describe the relationships among variables, the structure learning for a BN inference [18]. There are three kinds of BN structure learning algorithms: (1) constraint–based algorithms, using conditional independence tests to learn the dependency of variables from data; [19] (2) score–based algorithms, using score criteria as the objective functions. And (3) hybrid algorithms, combining the above two kinds of algorithms.

Score–based algorithms use the score function as the objective function, trying to find the DAG with the highest score (sometimes lowest score) using the search strategy

algorithm (for example the Max–min Hill–Climb algorithm (MMHC) [20], the hill–climbing algorithm, or the simulated annealing algorithm [21], genetic algorithm [22] and so on). There have been well–defined score functions such as Bayesian information criterion (BIC), Minimum Description Length score [23], and the Bayesian Gaussian equivalent (BGe) score [24], but because the Bayesian information criterion (BIC) has been one of the most widely used score–based algorithms and is used as the base algorithm for our method, a detailed description of it is given below.

Given a training data $D$, the BIC scoring function can be written as:

$$\text{BIC}(D) = \sum_{i=1}^{n} \sum_{k=1}^{r_i} \sum_{j=1}^{q_i} m_{ijk} \ln \frac{m_{ijk}}{m_{ij}} - \frac{1}{2}(\ln N) \sum_{i=1}^{n} q_i(r_i - 1) \tag{2}$$

where $m_{ijk}$ is the number of samples that $x_i = k$ and its parents are in their $j$-th configuration. Likewise, $m_{ij}$ is the number of samples in the dataset that variable is $i$-th and its parents are in their $j$-th configuration. $q_i$ is the number of possible configurations for parents of $i$-th random variable. $r_i$ is the number of different states of $i$-th random variable. $n$ is the number of random variables in BN. $N$ is the number of samples. In Equation (2), the first part is the model likelihood that is used to measure the fit of the DAG structure and data and the second part is a penalization of model complexity by assuming that every DAG shared the same probability. Intuitively, BIC selects the simplest model that fits with the data.

*2.3. Loss Functions*

Loss functions (LFs) are functions that map the value of a random event or its related random variable to a non–negative real number to represent the 'risk' or 'loss' of that random event [25]. It now has been introduced to engineering fields to estimate the loss caused by the deviations of variables [26].

Inverted normal loss function (INLF), shown in Equation (3) has been a widely used LF at present. The characteristic of INLF is that the loss will not increase indefinitely. When it reaches the predetermined threshold, the loss will reach the estimated maximum value and stop rising. Additionally, the value of INLF will not drop to a negative value [27]:

$$\text{INLF} : L(x) = EML \times \left[ 1 - exp\left( -(x - T)^2 / 2\gamma^2 \right) \right] \tag{3}$$

where $EML$ is the estimated maximum loss, which is determined by historical data or expert knowledge. $x$ is a variable whose deviation will cause some kind of loss, $T$ is its target value given by designing, $\gamma$ is the shape parameter and needs to be determined from additional information, i.e., expert knowledge.

## 3. A New Knowledge Guided iBWL Method

### 3.1. A New Inconsistent Knowledge Fusion Guided Score Function for BN Structure Learning

Generally, finding the optimal DAG with the highest score is NP–hard [28]. To tackle this problem, most existing approaches rely on heuristics to get the solution. In order to support the heuristic algorithm in obtaining better results, adding expert knowledge can greatly improve the performance of the algorithm [29].

However, expert knowledge reflects human understanding about a domain and the accuracy of the knowledge depends on how much the expert knows about the domain. Then, the major limitations of the existing approaches are two–fold: (1) the inconsistency among knowledge provided by different experts about the same issue has not been considered; and (2) the accuracy of the knowledge provided by a given expert has not been considered either. Therefore, in this paper, a new inconsistent knowledge fusion guided score function for BN structure learning was proposed based on the abovementioned BIC scoring method. In our approach, expert knowledge is no longer regarded as 100% definite nor 100% accurate but with probabilities. In addition, prior knowledge provided by more

than one expert was synchronously taken into account. Correspondingly, a fusion strategy of background knowledge provided by different experts was proposed.

There are *n* variables in a dataset, denoted as $N = \{ x_1, x_2, x_3, \ldots, x_n\}$. In our case, $x_i$ ($i = 1, 2, 3, \ldots, n$) are listed in Table 1. For any $x_i, x_j \in N$, there are three different types of expert knowledge about the relationship between $x_i$ and $x_j$, denoted as:

$$p(x_i \sim x_j) = \begin{cases} p(x_i \rightarrow x_j) \\ p(x_i \leftarrow x_j) \\ p(x_i \bowtie x_j) \end{cases} \tag{4}$$

where $p(x_i \rightarrow x_j)$ is the probability that $x_i$ has direct impact on $x_j$, $p(x_i \leftarrow x_j)$ is the probability that $x_j$ has direct impact on $x_i$, and $p(x_i \bowtie x_j)$ is the probability of no direct relationship between $x_i$ and $x_j$. Every expert needs to give their understanding on $C_n^2$ relationships, e.g., for three variables data, every expert needs to give $p(x_1 \sim x_2)$, $p(x_2 \sim x_3)$, $p(x_1 \sim x_3)$. For the detail of $p(x_i \sim x_j)$, there are four conditions.

1.  If an expert thinks the probability of $x_i$ having direct impact on $x_j$ is 50%, thus $p(x_i \rightarrow x_j)$ = 0.5;
2.  If an expert has no knowledge of the relationship between $x_i$ and $x_j$, then $p(x_i \rightarrow x_j) = p(x_i \leftarrow x_j) = p(x_i \bowtie x_j) = 1/3 \approx 0.333$;
3.  If an expert only gives one probability out of the three probabilities, then the remaining probabilities will be divided equally. For instance, if an expert believes $p(x_i \rightarrow x_j) = 0.4$, but has no idea about $p(x_j \rightarrow x_i)$ nor $p(x_i \bowtie x_j)$, then $p(x_j \rightarrow x_i) = p(x_i \bowtie x_j) = (1 - 0.4)/2 = 0.3$;
4.  An expert only needs to give two probabilities out of the three probabilities, because the sum of the three probabilities is 1. For instance, an expert believes $p(x_i \rightarrow x_j) = 0.4$, $p(x_i \bowtie x_j) = 0.3$, then $p(x_i \leftarrow x_j) = 1 - 0.4 - 0.3 = 0.3$;
5.  Since every type of knowledge is mutually exclusive, the relationship between $x_i$ and $x_j$ with direct knowledge is quantified with $p(x_i \rightarrow x_j) + p(x_i \leftarrow x_j)$. For example, if an expert believes that the probability of $x_i$ and $x_j$ having a direct relation is 0.6, thus $p(x_i \rightarrow x_j) = 0.6/2$, $p(x_i \leftarrow x_j) = 0.6/2$.

After consulting the *k*-th expert ($k = 1, 2, \ldots, m$. *m* is the number of experts), his/her knowledge is placed into one matrix, called the knowledge matrix of the *k*-th expert, noted as $\varepsilon^k$. The *i*-th row and *j*-th column element in $\varepsilon^k$ represents $p(x_i \rightarrow x_j)$ and the diagonal elements in the matrix are set as 0 since there is no relationship between the variable and itself. Figure 2a is an example of a knowledge matrix. In this matrix, the $p(x_1 \rightarrow x_2) = 0.5$, and $p(x_2 \rightarrow x_1) = 0.1$.

To deal with the inconsistency of the knowledge among different experts, the information fusion strategy based on expert confidence was proposed. For the knowledge given by the *k*-th expert, a confidence coefficient $c_k$ is assigned according to the reliability of his/her experience (comprehensively considering the working years, educational experience, devotional years on an issue and so on). Indeed, the quantification of experience is difficult, and there has not been a widely accepted way to perform it. It is a big challenge worthy of more attention and research.

The corresponding knowledge matrix $\varepsilon^k$ is obtained by consulting the *k*-th expert, then a fused knowledge matrix given by *m* experts is denoted as $\varepsilon$, and it can be expressed as:

$$\varepsilon = \sum_{k=1}^{m} c_k \varepsilon^k \bigg/ \sum_{k=1}^{m} c_k \tag{5}$$

After the fused matrix is obtained through the above description, the next step is to formulate the score function.

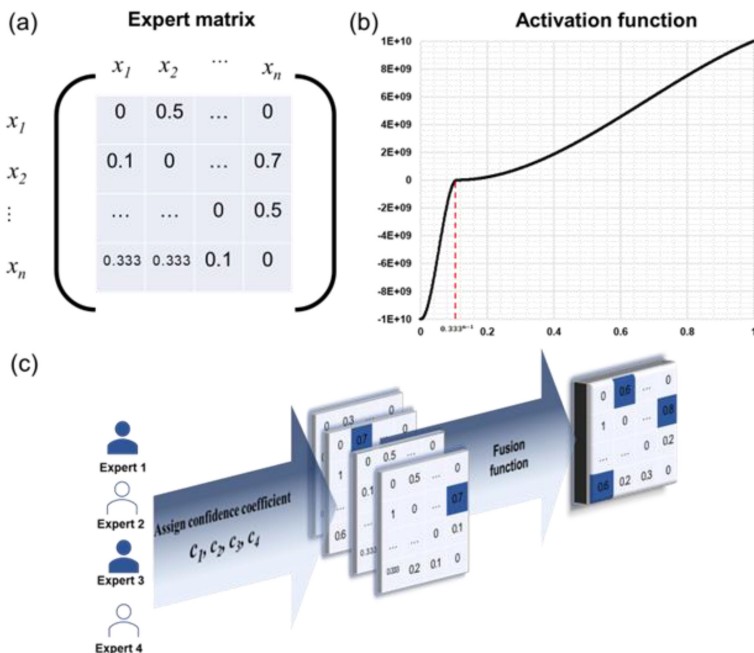

**Figure 2.** (**a**) Expert knowledge matrix, the value at *i*-th row, *j*-th column refers $p(x_i{\rightarrow}x_j)$ the probability of the *i*-th variable having a direct impact on the *j*-th variable. (**b**) Activation function, an example of cubic interpolation function, in which positive infinity is replaced by the real number $1 \times 10^{10}$. (**c**) The flow chart demonstrates how to obtain the fused knowledge matrix. In the first step, several experts are invited to give their personal opinions on the relationships of each variable pair, then they are assigned a confidence coefficient according to their experience. In the second step, serval knowledge matrixes are fused together to obtain one fused knowledge matrix according to the fusion function.

Bayesian information criterion (BIC) is the most commonly used score function and is also known as the Minimum Description Length, which is expressed as Equation (2). The first part of it is the likelihood, and the second part is a prior assuming every DAG shares the uniform probability. In order to integrate knowledge, the second part should be modified and based on this, a new inconsistent knowledge fusion guided score function (Score$_{ikf}$) is written as:

$$\text{Score}_{ikf} = \sum_{i=1}^{n} \sum_{k=1}^{r_i} \sum_{j=1}^{q_i} m_{ijk} ln \frac{m_{ijk}}{m_{ij}} + logP(G) \tag{6}$$

The first part of Score$_{ikf}$ is the same as a BIC score function, namely, it is the likelihood, and our method of integrating expert information is to estimate $logP(G)$. Because the first summation symbol of the likelihood is summing over each node of the $G$, thus it can be decomposed into every node. The prior also has this characteristic. For one node $x_i$ in $G$, the prior is defined as:

$$\log P(x_i) = f \left[ \sum_{j=1}^{n} \varepsilon_{ij} \right] \tag{7}$$

where $f$ is the activation function.

To make sure the algorithm is robust to noise but sensitive to expert most confident knowledge (especially to the 100% confident knowledge), the activation function $f$ should have the following characteristics:

1. For a given threshold value $\tau$, if $\prod\limits_{j=1, j\neq i}^{n} \varepsilon_{ij} \leq \tau$, this means no useful expert knowledge about the $i$-th variable is available, then $\log P(x_i) = f[\tau] = 1$.

2. The activation function should be as smooth as possible under the threshold value $\tau$ to ensure that slight random noise will not cause drastic changes in the scoring function to enhance the robustness of the algorithm. $\tau$ can be set according to the knowledge or be optimized by genetic algorithms and so on. In our case, $\tau = (1/3)^{n-1}$.

3. If all experts are 100% confident about the relationship between $x_i$ and $x_j$, which means $\prod\limits_{j=1, j\neq i}^{n} \varepsilon_{ij} = 1$, then $f[1] = \infty$ (or a very big positive number) to make sure the relationship learned from the data makes no difference.

There are many options for activation functions. Figure 2b is an example of using the cubic spline interpolation as the activation function and the interpolation conditions are based on the information entropy. The interpolation conditions of it are expressed in (8).

$$
\begin{cases}
\left[ P(x_i) \leq (1/3)^{n-1} \right] = 0 \\
f'\left[ P(x_i) \leq (1/3)^{n-1} \right] = 0 \\
\qquad\qquad f[1] = \infty \\
\qquad\qquad f'[1] = \infty
\end{cases}
\tag{8}
$$

where $f'[\cdot]$ is the first derivative of $f[\cdot]$.

In summary, given a certain dataset $D$ and the fused knowledge matrix, $\varepsilon$, the inconsistent knowledge fusion guided function score of the Bayesian network algorithm should be:

$$
\begin{aligned}
\log P(G|D) &= \sum_{i=1}^{n} \log P(D|x_i) + \sum_{i=1}^{n} \log P(x_i) \\
&= \sum_{i=1}^{n} \sum_{k=1}^{r_i} \sum_{j=1}^{q_i} m_{ijk} \ln \frac{m_{ijk}}{m_{ij}} + \sum_{i=1}^{n} f\left[ \sum_{j=1}^{n} \varepsilon_{ij} \right]
\end{aligned}
\tag{9}
$$

This score is applied in the theoretical framework based on a scoring search algorithm. The scoring search algorithm we selected was a hill–climbing algorithm whose pseudo code is shown in Algorithm 1.

---

**Algorithm 1** Hill–climbing algorithm.

| | |
|---|---|
| 1: | **Input** Observed data D; score function f; maximum iteration times NumIter; restart times NumStart; |
| 2: | G is an empty DAG, |
| 3: | ResultG = G; |
| 4: | **for** r **from** 1 **to** NumStart: |
| 5: | **for** n **from** 1 **to** NumIter: |
| 6: |   legal operation is one of the operations that adding, deleting, or flipping edge on DAG at the same time the DAG remains acyclic; |
| 7: |    find a legal operation that maximizes f(G\*, D, K) – f(G, D, K), where G\* is G after one legal operation; |
| 8: |     **if** f(G\*, D) – f(G, D) > 0: |
| 9: |      G = G\*; |
| 10: |     **else**: |
| 11: |      break; |
| 12: |   **if** f(G, D) – f(ResultG, D) > 0: |
| 13: |   ResultG = G; |
| 14: | **return** ResultG; |

---

### 3.2. Weighted Loss Function Model for Reliability Evaluation

As we mentioned above, loss function (LF) is a kind of function that estimates the 'risk' or 'loss' caused by the deviations of variables from the corresponding expected values. It also

can be used to estimate the loss of the reliability caused by the deviations of variables from the corresponding expected values in industrial fields if it is described as Equation (10):

$$R(x) = 100 \times \exp\left(-\sum_{i=1}^{n} (x_i - T_i)^2 / 2\gamma_i^2\right) \tag{10}$$

where $x_i$ ($i = 1, \ldots, n$) is the $i$-th variable whose fluctuation affects the system reliability, $T_i$ is its expected value, and $\gamma_i$ is its shape parameter.

However, according to engineering experience and knowledge, different variables have different importance to the system reliability. For example, in our case, the Bolt preload (*LS_YJL*) has a much bigger influence on the sealing performance than the inner diameter of the head (*SR2*) does. Therefore, it is more reasonable to take the importance of variables into account. Correspondingly, a new weighted reliability score was proposed as:

$$R(D) = 100 \times \exp\left(-\sum_{i=1}^{n} k_i (x_i - T_i)^2 / 2\gamma_i^2\right) \tag{11}$$

where $k_i$ is the quantified important index of $x_i$.

There are many ways to quantify the importance of a variable [30]. In our case, the degree centrality was used because it can describe how many other variables are affecting or being affected by a given variable.

Degree centrality is defined as the number of links incidents upon a node/variable [31]. In a directed network (where edges have direction), it is usually defined as the sum of the indegree and the outdegree. Indegree is a count of the number of edges directed to the node/variable and outdegree is the number of edges that the node/variable directs to others. Hence the importance index of $x_i$ is defined as:

$$k_i = \deg(x_i) / \sum_{j=1}^{j=n} \deg(x_j) \tag{12}$$

where $deg\,(x_i)$ is the degree centrality of $x_i$, $m$ is the amount of selected key nodes, and $k_i$ describes the importance of $x_i$ in the whole network.

## 4. Results and Discussion

Five experienced experts (one professor, two associate professors and two engineers) were invited to provide background knowledge about the relationships among variables of the RPV sealing system. All these experts had been working on RPV design or analysis for at least three years. The confidence coefficients of knowledge provided by the experts are assigned in Table 2. Compared with professional title, working experience is more important. In our case, because of the speciality of RPV, it was not possible to have a plentiful number of professionals available. The number of experts in our study was only five, and their confidence coefficients were only divided into five levels. For other cases, a different ranking method may be better.

**Table 2.** The confidence coefficients for the knowledge provided by experts.

| Expert | Professional Title | Working Years in the Related Area | Working Years in RPV Design and Analysis | Confidence Coefficient |
|--------|--------------------|-----------------------------------|------------------------------------------|------------------------|
| E1 | Professor | 25 | 10 | 5 |
| E2 | Associate professor A | 10 | 8 | 4 |
| E3 | Associate professor B | 8 | 8 | 4 |
| E4 | Engineer A | 6 | 3 | 2 |
| E5 | Engineer B | 5 | 3 | 2 |

The fused expert knowledge matrix using the fusion strategy mentioned before is shown in Figure 3. Based on the fused expert knowledge, a BN was obtained by the iBWL method. To verify the effectiveness of expert knowledge, the regular BIC learning algorithm was also performed on the same data. The BNs obtained by both methods are shown in Figure 4. Figure 4a shows the BN obtained by our iBWL method. It is obvious to see that:

1. Bolt preload (*LS_YJL*) affects the compression of the gasket. It is an important variable to ensure sealing performance, which will affect *JX_IN, JX_OUT, ZJ_U, ZJ_D, FL_ZKL* and other variables [3,32], but it is independent with the structure variables. For a FEA simulation model, *LS_YJL* is an input variable. FEA could not calculate the values of it nor the influence of it to the RPV sealing system. This is the reason that BN or other machine learning methods are needed for this issue.

2. Displacement variables (*ZX_IN, ZX_OUT, ZJ_D, ZJ_U, FL_ZKL*) play important roles in the sealing system [33]. They are mainly affected by *LS_YJL*. In the network topology structure, there is no displacement variable point to the structure variables, which is completely consistent with the physics.

3. The radial separation of the gasket is harmful and will lead to a bending moment or shear force. The too–big radial separation will result in gasket premature failure, but compared to the axial separation, the radial separation is less important. The sealing performance will seriously descend while the axial separation of the gasket would be larger than expected [33], with the axial separation represented by *ZX_IN, ZX_OUT*. According to Figure 4a, only the ramp angle has a direct effect on the radial displacement, while other parameters do not affect it. This represents a less important role of the radial displacement variable than other displacement variables, which is consistent with expert knowledge.

4. The size variables in the sealing area (*D11, D12, D13, D14, D15, θ*), shown in Figure 1, interact with each other and are related to some of the other dimension variables. *D11, D14, D15* are prominent in such variables

| | FL_JXWYL | LS_YJL | FL_ZKL | ZJ_D | ZJ_U | JX_OUT | JX_IN | ZX_OUT | ZX_IN | θ | T2 | T1 | H3 | H2 | H1 | SR2 | D17 | D15 | D14 | D13 | D12 | D11 | D10 | D9 | D8 | D7 | D3 | D2 | D1 |
|---|---|---|---|---|---|---|---|---|---|---|---|---|---|---|---|---|---|---|---|---|---|---|---|---|---|---|---|---|---|
| FL_JXWYL | 0 | 0 | 0 | 0 | 0 | 0 | 0 | 0 | 0 | 0 | 0 | 0 | 0 | 0 | 0 | 0 | 0 | 0 | 0 | 0 | 0 | 0 | 0 | 0 | 0 | 0 | 0 | 0 | 0 |
| LS_YJL | 0.1 | 0 | 0.9 | 0.9 | 0.9 | 0.1 | 0.1 | 0.9 | 0.9 | 0 | 0 | 0 | 0 | 0 | 0 | 0 | 0 | 0 | 0 | 0 | 0 | 0 | 0 | 0 | 0 | 0 | 0 | 0 | 0 |
| FL_ZKL | 0 | 0 | 0 | 0 | 0 | 0 | 0 | 0 | 0 | 0 | 0 | 0 | 0 | 0 | 0 | 0 | 0 | 0 | 0 | 0 | 0 | 0 | 0 | 0 | 0 | 0 | 0 | 0 | 0 |
| ZJ_D | 0 | 0 | 0 | 0 | 0 | 0 | 0 | 0 | 0 | 0 | 0 | 0 | 0 | 0 | 0 | 0 | 0 | 0 | 0 | 0 | 0 | 0 | 0 | 0 | 0 | 0 | 0 | 0 | 0 |
| ZJ_U | 0 | 0 | 0 | 0 | 0 | 0 | 0 | 0 | 0 | 0 | 0 | 0 | 0 | 0 | 0 | 0 | 0 | 0 | 0 | 0 | 0 | 0 | 0 | 0 | 0 | 0 | 0 | 0 | 0 |
| JX_OUT | 0.333 | 0 | 0.333 | 0.333 | 0.333 | 0 | 0.333 | 0.333 | 0.333 | 0 | 0 | 0 | 0 | 0 | 0 | 0 | 0 | 0 | 0 | 0 | 0 | 0 | 0 | 0 | 0 | 0 | 0 | 0 | 0 |
| JX_IN | 0.333 | 0 | 0.333 | 0.333 | 0.333 | 0.333 | 0 | 0.333 | 0.333 | 0 | 0 | 0 | 0 | 0 | 0 | 0 | 0 | 0 | 0 | 0 | 0 | 0 | 0 | 0 | 0 | 0 | 0 | 0 | 0 |
| ZX_OUT | 0.333 | 0 | 0.333 | 0.333 | 0.333 | 0.333 | 0.333 | 0 | 0.333 | 0 | 0 | 0 | 0 | 0 | 0 | 0 | 0 | 0 | 0 | 0 | 0 | 0 | 0 | 0 | 0 | 0 | 0 | 0 | 0 |
| ZX_IN | 0.333 | 0 | 0.333 | 0.333 | 0.333 | 0.333 | 0.333 | 0.333 | 0 | 0 | 0 | 0 | 0 | 0 | 0 | 0 | 0 | 0 | 0 | 0 | 0 | 0 | 0 | 0 | 0 | 0 | 0 | 0 | 0 |
| θ | 0.3 | 0 | 0.7 | 0.7 | 0.7 | 0.7 | 0.7 | 0.7 | 0.7 | 0 | 0 | 0 | 0 | 0 | 0 | 0 | 0.333 | 0.333 | 0.333 | 0.333 | 0.333 | 0.1 | 0.1 | 0.1 | 0.1 | 0.1 | 0.1 | 0.1 | 0.1 |
| T2 | 0.333 | 0 | 0.333 | 0.333 | 0.333 | 0.333 | 0.333 | 0.333 | 0.333 | 0.333 | 0 | 0.333 | 0.333 | 0.333 | 0.333 | 0.333 | 0.333 | 0.333 | 0.333 | 0.333 | 0.333 | 0.1 | 0.333 | 0.333 | 0.333 | 0.333 | 0.333 | 0.333 | 0.333 |
| T1 | 0.333 | 0 | 0.333 | 0.333 | 0.333 | 0.333 | 0.333 | 0.333 | 0.333 | 0.333 | 0.333 | 0 | 0.333 | 0.333 | 0.333 | 0.333 | 0.333 | 0.333 | 0.333 | 0.333 | 0.333 | 0.1 | 0.333 | 0.333 | 0.333 | 0.333 | 0.333 | 0.333 | 0.333 |
| H3 | 0.333 | 0 | 0.333 | 0.333 | 0.333 | 0.333 | 0.333 | 0.333 | 0.333 | 0.333 | 0.333 | 0.333 | 0 | 0.333 | 0.333 | 0.333 | 0.333 | 0.333 | 0.333 | 0.333 | 0.333 | 0.1 | 0.333 | 0.333 | 0.333 | 0.333 | 0.333 | 0.333 | 0.333 |
| H2 | 0.333 | 0 | 0.333 | 0.333 | 0.333 | 0.333 | 0.333 | 0.333 | 0.333 | 0.333 | 0.333 | 0.333 | 0.333 | 0 | 0.333 | 0.333 | 0.333 | 0.333 | 0.333 | 0.333 | 0.333 | 0.1 | 0.333 | 0.333 | 0.333 | 0.333 | 0.333 | 0.333 | 0.333 |
| H1 | 0.333 | 0 | 0.333 | 0.333 | 0.333 | 0.333 | 0.333 | 0.333 | 0.333 | 0.333 | 0.333 | 0.333 | 0.333 | 0.333 | 0 | 0.333 | 0.333 | 0.333 | 0.333 | 0.333 | 0.333 | 0.1 | 0.333 | 0.333 | 0.333 | 0.333 | 0.333 | 0.333 | 0.333 |
| SR2 | 0.333 | 0 | 0.1 | 0.1 | 0.1 | 0.1 | 0.1 | 0.1 | 0.1 | 0.1 | 0.1 | 0.1 | 0.1 | 0.1 | 0.1 | 0 | 0.1 | 0.1 | 0.1 | 0.1 | 0.1 | 0.1 | 0.1 | 0.1 | 0.1 | 0.1 | 0.1 | 0.1 | 0.1 |
| D17 | 0.333 | 0 | 0.1 | 0.1 | 0.1 | 0.1 | 0.1 | 0.1 | 0.1 | 0.1 | 0.1 | 0.1 | 0.1 | 0.1 | 0.1 | 0.1 | 0 | 0.333 | 0.333 | 0.333 | 0.333 | 0.333 | 0 | 0 | 0 | 0 | 0 | 0 | 0 |
| D15 | 0.333 | 0 | 0.5 | 0.5 | 0.5 | 0.333 | 0.333 | 0.5 | 0.5 | 0.333 | 0.333 | 0.333 | 0.333 | 0.333 | 0.333 | 0.333 | 0.333 | 0 | 0.5 | 0.333 | 0.5 | 0.333 | 0.333 | 0.333 | 0.333 | 0.333 | 0.333 | 0.333 | 0.333 |
| D14 | 0.333 | 0 | 0.5 | 0.5 | 0.5 | 0.333 | 0.333 | 0.5 | 0.5 | 0.333 | 0.333 | 0.333 | 0.333 | 0.333 | 0.333 | 0.333 | 0.333 | 0.5 | 0 | 0.333 | 0.333 | 0.333 | 0.1 | 0.1 | 0.1 | 0.1 | 0.1 | 0.1 | 0.1 |
| D13 | 0.1 | 0 | 0.5 | 0.5 | 0.5 | 0.1 | 0.1 | 0.5 | 0.5 | 0.333 | 0.333 | 0.333 | 0.333 | 0.333 | 0.333 | 0.333 | 0.333 | 0.333 | 0.333 | 0 | 0.333 | 0.333 | 0.333 | 0.333 | 0.333 | 0.333 | 0.333 | 0.333 | 0.333 |
| D12 | 0.333 | 0 | 0.5 | 0.5 | 0.5 | 0.333 | 0.333 | 0.5 | 0.5 | 0.333 | 0.333 | 0.333 | 0.333 | 0.333 | 0.333 | 0.333 | 0.333 | 0.333 | 0.333 | 0.333 | 0 | 0.5 | 0.333 | 0.333 | 0.333 | 0.333 | 0.333 | 0.333 | 0.333 |
| D11 | 0.1 | 0 | 0.5 | 0.5 | 0.5 | 0.1 | 0.1 | 0.5 | 0.5 | 0.333 | 0.333 | 0.333 | 0.333 | 0.333 | 0.333 | 0.333 | 0.333 | 0.333 | 0.333 | 0.333 | 0.5 | 0 | 0.333 | 0.333 | 0.333 | 0.333 | 0.333 | 0.333 | 0.333 |
| D10 | 0.333 | 0 | 0.333 | 0.333 | 0.333 | 0.333 | 0.333 | 0.333 | 0.333 | 0.1 | 0.333 | 0.333 | 0.333 | 0.333 | 0.333 | 0.333 | 0 | 0.333 | 0.1 | 0.333 | 0.333 | 0.333 | 0 | 0.333 | 0.333 | 0.333 | 0.333 | 0.333 | 0.333 |
| D9 | 0.333 | 0 | 0.333 | 0.333 | 0.333 | 0.333 | 0.333 | 0.333 | 0.333 | 0.1 | 0.333 | 0.333 | 0.333 | 0.333 | 0.333 | 0.333 | 0 | 0.333 | 0.1 | 0.333 | 0.333 | 0.333 | 0.333 | 0 | 0.333 | 0.333 | 0.333 | 0.333 | 0.333 |
| D8 | 0.333 | 0 | 0.333 | 0.333 | 0.333 | 0.333 | 0.333 | 0.333 | 0.333 | 0.1 | 0.333 | 0.333 | 0.333 | 0.333 | 0.333 | 0.333 | 0 | 0.333 | 0.1 | 0.333 | 0.333 | 0.333 | 0.333 | 0.333 | 0 | 0.333 | 0.333 | 0.333 | 0.333 |
| D7 | 0.333 | 0 | 0.333 | 0.333 | 0.333 | 0.333 | 0.333 | 0.333 | 0.333 | 0.1 | 0.333 | 0.333 | 0.333 | 0.333 | 0.333 | 0.333 | 0 | 0.333 | 0.1 | 0.333 | 0.333 | 0.333 | 0.333 | 0.333 | 0.333 | 0 | 0.333 | 0.333 | 0.333 |
| D3 | 0.333 | 0 | 0.333 | 0.333 | 0.333 | 0.333 | 0.333 | 0.333 | 0.333 | 0.1 | 0.333 | 0.333 | 0.333 | 0.333 | 0.333 | 0.333 | 0 | 0.333 | 0.1 | 0.333 | 0.333 | 0.333 | 0.333 | 0.333 | 0.333 | 0.333 | 0 | 0.333 | 0.333 |
| D2 | 0.333 | 0 | 0.333 | 0.333 | 0.333 | 0.333 | 0.333 | 0.333 | 0.333 | 0.1 | 0.333 | 0.333 | 0.333 | 0.333 | 0.333 | 0.333 | 0 | 0.333 | 0.1 | 0.333 | 0.333 | 0.333 | 0.333 | 0.333 | 0.333 | 0.333 | 0.333 | 0 | 0.333 |
| D1 | 0.333 | 0 | 0.333 | 0.333 | 0.333 | 0.333 | 0.333 | 0.333 | 0.333 | 0.1 | 0.333 | 0.333 | 0.333 | 0.333 | 0.333 | 0.333 | 0 | 0.333 | 0.1 | 0.333 | 0.333 | 0.333 | 0.333 | 0.333 | 0.333 | 0.333 | 0.333 | 0.333 | 0 |

**Figure 3.** The fused expert knowledge matrix for RPV sealing system.

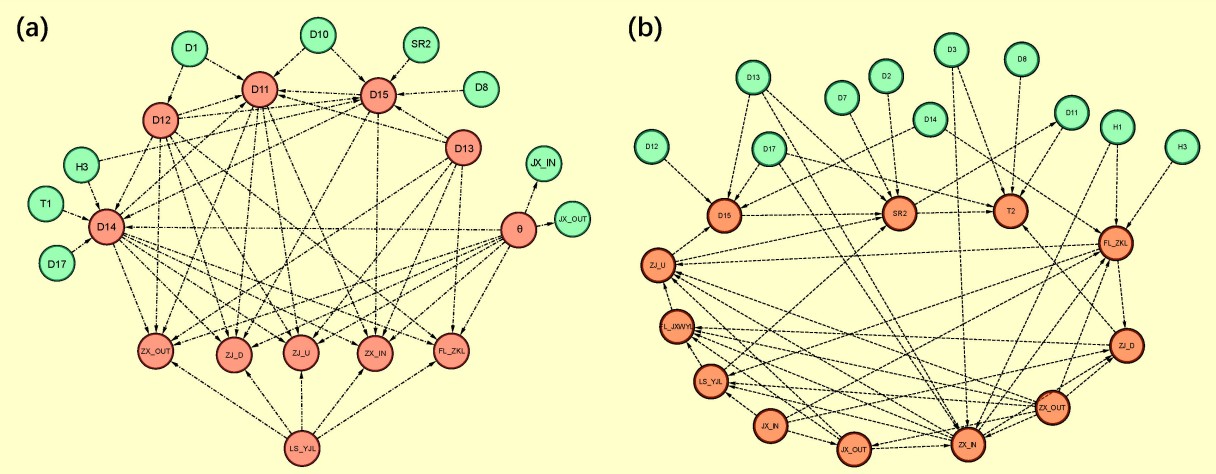

**Figure 4.** Bayesian network structures are learned by two different methods. According to (15), the nodes/variables are marked in orange, if ki > 4. (**a**) The Bayesian network is learned by the iBWL method. (**b**) The Bayesian network is learned by the regular BIC learning algorithm.

According to the network learned by the regular BIC score which is shown in Figure 4b, there were several inconsistencies with the physical mechanism or expert knowledge:

1. According to the mechanism knowledge, *LS_YJL* is independent of *SR2*, the size of the spherical head, but the *LS_YJL* has a direct edge to *SR2* in Figure 4b.
2. According to design drawings in Figure 1b, *D15* and *D13* determine the sealing rings position simultaneously, thus *D15* should be very sensitive to *D13*, but there is no direct nor indirect connection between *D15* and *D13* in Figure 4b.
3. *D13* determines the assembly position of the gasket. The gasket should be at the position shown in Figure 1b, with three sides in contact with the surface, so that the sealing ring has a higher constraint to ensure the sealing performance. *D12* and *D14* determine the position of the sealing groove while reasonable positions of sealing grooves ensure the sealing ring has good sealing performance. If the distance between the two sealing grooves is too close, the sealing performance of a single sealing ring will be weakened. And the sealing performance will deteriorate when the distance is too long [34]. All in all, *D11*, *D12*, *D13*, and *D14* are also important variables affecting the displacement parameters, which were not learned by the BIC method.
4. *ZX_IN* and *ZX_OUT* are two interrelated displacement parameters, thus, they should have similar connections, while in Figure 4b, there are more directed edges pointed to *ZX_IN* than *ZX_OUT*. This is not consistent with the experts' expectations.

By comparison, it can be found that the relationships among the variables captured by the network topology structure using our inconsistent knowledge fusion method is more accurate according to physical cognition. The expert knowledge integrated BN not only excavates the influence relationship between the variables from the data but also effectively considers the priceless knowledge accumulated by experts. More importantly, it considers the inconsistency and reliability of the knowledge provided by different experts. This is an effective method to describe the dependence among variables to identify key features, especially when the project requires high reliability and experimental data is scarce.

According to Equation (12), we obtain the centrality degree $k_i$ of each variable and identified *D14, D15, D11, D12, θ, D13, ZX_OUT, ZJ_D, ZJ_U, ZX_IN, FL_ZKL, LS_YJL* as key variables (their $k_i$ are shown in Table 3).

**Table 3.** Importance index of key variables.

| Node/Variable | Centrality Degree | $k_i$ | Node/Variable | Centrality Degree | $k_i$ |
|---|---|---|---|---|---|
| D14 | 12 | 0.136 | ZX_OUT | 6 | 0.068 |
| D15 | 10 | 0.114 | ZJ_D | 6 | 0.068 |
| D11 | 10 | 0.114 | ZJ_U | 6 | 0.068 |
| D12 | 8 | 0.091 | ZX_IN | 6 | 0.068 |
| θ | 8 | 0.091 | FL_ZKL | 5 | 0.057 |
| D13 | 6 | 0.068 | LS_YJL | 5 | 0.057 |

Then, using the weighted loss function shown in Equation (11), the reliabilities of all samples were quantified and shown in Figure 5a. For the reliability function, $\gamma = 1$ since samples were preprocessed with mean normalization. The designed parameter values of the RPV of the Liaoning Hongyanhe Nuclear Power Plant were used as the target values.

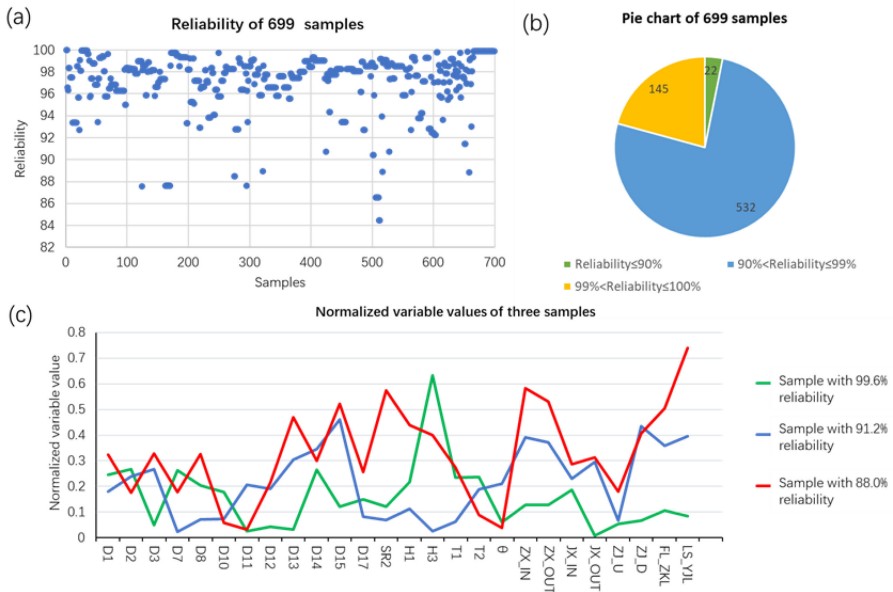

**Figure 5.** (**a**) Reliability scatter diagram of 699 samples. (**b**) Pie chart of reliability of 699 samples. (**c**) The normalized variable values of three typical samples with different reliabilities.

To active the generalization, robustness of the BN model and to test the performance of our iBWL method, several simulation samples were designed with unacceptable structure values or status values on purpose. Figure 5a,b also shows the distribution of reliabilities of them. It is obvious that as what we expected, the samples whose designed variables' value have big variations by us on purpose, did not have good enough reliability scores.

There were almost no deviations of the variables of samples whose reliability scores were higher than 99.00%. Therefore, three typical samples, whose reliabilities were 99.6%, 91.2% and 88.0%, respectively, were chosen as examples. Figure 5c shows the variations of their variables. From Figure 5c we can see that:

1.  For the green sample, *H3* has the biggest deviation from its target value and the deviations of other variables are comparatively small, close to 0. According to Table 3, H3 is not a key variable, therefore the deviation of it did not change the sealing performance very much. The reliability of it is still 99.6%.

2.  For the blue sample with 91.2% reliability, although the deviations of *D14*, *D15*, *ZX_IN*, *ZX_OUT*, *ZJ_D*, *FL_ZKL*, *LS_YJL* are not as big as that of *H3* in the green sample, they are more important variables according to Table 3. Consequently, the blue sample has lower reliability than the green sample does.

3.　For the red sample with 88.0% reliability, *D10*, *D11*, *T2*, and $\theta$ have the highest deviations. They are important structure variables and their deviations from their corresponding expected values led to a dramatic decline in sealing reliability.

These three samples proved that not only the deviations of variables but also their importance is taken into consideration in our new iBWL method.

For a FEA simulation model, *LS_YJL* is an input variable. FEA could not calculate the values of it nor the influence of it to the RPV sealing system. For the displacement variables (*ZX_IN*, *ZX_OUT*, *ZJ_D*, *ZJ_U*, *FL_ZKL*) that play important roles in the sealing system, FEA can only calculate the values of them under certain circumstances but cannot evaluate their influence on the RPV sealing system either.

For the reliability analysis model of a nuclear reactor pressure vessel, it is difficult to use statistical methods for verification, because it is impossible to obtain a large number of design schemes and real reactor reliability data. To verify the accuracy of the iBWL method, we applied it on two RPVs that are serving in China. The data of these two RPVs did not participate in the previous model learning.

The Fuqing nuclear power plant is located in Fuqing City, Fujian Province, China. The planned installed capacity is 6 million KW PWR nuclear power units. Units 1 to 4 use CNNC's cpr–1000 reactor, and units 5 and 6 use CNNC's Hualong 1 (formerly known as ACP–1000) reactor. Using the hydraulic pressure test data of Fuqing 4 and 5 units, the reliability of the equipment was analyzed. The reactor pressure vessel hydraulic pressure test mainly included a strain test, main bolt load test and deformation test. The strain test and deformation test related to the sealing performance of reactor pressure vessel and mainly included the strain at the head flange and vessel flange, the corner of the head flange and vessel flange, and the axial and radial displacement between the head and vessel flange. Through the above tests, the values of key variables were obtained and are listed in Table 4.

**Table 4.** Key variable values after normalization of Fuqing 4, 5 units.

| Node/Variable | Fuqing 4 Unit | Fuqing 5 Unit | Node/Variable | Fuqing 4 Unit | Fuqing 5 Unit |
|---|---|---|---|---|---|
| D14 | 0.565022422 | 0.538116592 | ZX_OUT | 0.71517225 | 0.654898238 |
| D15 | 0.372469636 | 0.331983806 | ZJ_D | 0.58882819 | 0.753666217 |
| D11 | 0.393382353 | 0.797794118 | ZJ_U | 0.79986268 | 0.537599129 |
| D12 | 0.250996016 | 0.398406375 | ZX_IN | 0.700518179 | 0.623336078 |
| $\theta$ | 0.665236052 | 0.25751073 | FL_ZKL | 0.705600748 | 0.675037485 |
| D13 | 0.398104265 | 0.44549763 | LS_YJL | 0.823301528 | 0.830323161 |

The values of the key variables were input into the weighted loss function shown in Equation (11), which is the second part of the iBWL method. The reliability of these two RPVs can therefore be obtained. The reliability scores of Fuqing 4 and 5 units were 99.310% and 99.661%, respectively. The RPVs of Fuqing 4 and 5 units both obtained 'high reliability' results, which are in line with expectations because they were running safely. These two real data verify that the iBWL method proposed in this paper can correctly evaluate the reliability of a RPV sealing system. Of course, we also admit that this verification method is arbitrary, as it is limited by the scarcity of data and the lack of fault data.

## 5. Conclusions

A new iBWL reliability estimation method was proposed for RPV sealing performance. To overcome the inherent shortcomings of Bayesian network structure learning, we proposed a novel knowledge integrated BN learning algorithm to fuse the information from data and inconsistent knowledge from multi–experts. The fusion strategy and activation function were proposed to overcome the inconsistency, to consider the reliability of the knowledge provided by different experts, and to improve the accuracy and the robustness of the algorithm. Based on the analysis of the learned BN topology structure, the key variables are identified. The weighted loss function model, then took key variables as input

to quantify the loss of sealing performance reliability caused by the deviations of these variables from their expected values. The case studies on simulated samples based on the RPV of Liaoning Hongyanhe Nuclear Power Plant and real data on other RPV in service in China are conducted to verify our method. The comparisons between the results obtained by our method and the regular BN method strongly proved the iBWL method's advantage in integrating both data and expert knowledge.

By comprehensively considering both the structure variables and status variables under different internal temperatures and pressures of a given RPV, our method is a powerful method to estimate the reliability of RPV sealing performance under different internal temperatures and pressures, thus, it can help with daily operation and maintenance. It also laid a valuable foundation for the realization of safety online monitoring and management if the corresponding measure methods are available. In addition, the key variables identified by the iBWL method are also helpful with the designing, analyzing and maintenance of RPV intrinsic safety. Future work should therefore include follow–up work designed to integrate more knowledge including literature and design cases to the BN learning algorithm and should be designed to optimize the quantification method of the confidence coefficient of experts.

**Author Contributions:** Investigation, H.H., C.L., Y.D., Z.Z. and X.W.; methodology, K.S.; project administration, X.C.; software, H.H.; supervision, K.S.; writing—original draft, H.H.; writing—review and editing, Y.L. and K.S. All authors have read and agreed to the published version of the manuscript.

**Funding:** This research was funded by Science and Technology on Reactor System Design Technology Laboratory grant number HT–KFKT–02–2020015.

**Institutional Review Board Statement:** Not applicable.

**Informed Consent Statement:** Not applicable.

**Data Availability Statement:** Not applicable.

**Acknowledgments:** The authors gratefully acknowledge the financial support for this work from Science and Technology on Reactor System Design Technology Laboratory (HT–KFKT–02–2020015).

**Conflicts of Interest:** The authors declare that they have no known competing financial interest or personal relationship that could have appeared to influence the work reported in this paper.

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
