# Peer review of "RPV Sealing Reliability Estimating Using a New Inconsistent Knowledge Fused Bayesian Network and Weighted Loss Function"

_processes, doi:10.3390/pr10061099_

Round 1
Reviewer 1 Report
1. I suggest an english review of the paper.
2. I am have doubt about the confidence coefficients for the knowledge provided by experts. How much does the cofidence coefficient for the professional title affect the results? The engineer A and B are only focus on this type of analysis? It is really difficult to quantify experience.

Reviewer 2 Report
This paper carries interesting novelty. This paper written very well all tables and graph presented in a better frame work. In conclusion section author should add more advantages of their work in daily life. Also little bit discussion about future work is needed.
Reviewer 3 Report
In this paper, a new inconsistent knowledge fused Bayesian Network and weighted loss function (iBWL) method is proposed to estimate the reliability of nuclear reactor pressure vessel (RPV) sealing performance under different internal temperatures and pressure. The paper is interesting, I recommend its publication if the following comments are addressed:
- The differences between the current study and previous studies need to be presented properly by accurately providing the appropriate description of the research gap or issues encountered in the literature. The Introduction section should provide a clear and precise statement of the motivation of the paper. It should also answer certain questions like why it is essential to propose a new method. What are the limitations associated with the already available methods in the literature? The introduction can be improved by addressing the main feature of the work; more explanation of the cited references with a highlight on the differences.
- The Literature Survey could be greatly improved. The author first needs to make comparisons of them and then draw the motivation of the paper. Neither the comparison of references and this work nor the corresponding conclusion is made in the paper. Thus, it is difficult for me to know the novelty and advantages of this paper over other works.
- The authors should point out the major contributions of this paper by using 3 to 5 brief bullet points at the end of the Introduction Section, right before the last paragraph.
- Why you didn’t consider a type of the probability of direct relationship between xi and xj in Eq. 4?
- Although the proposed method seems interesting, the structure of the Methodology Section is not presented in such a way that a general description of the method is given without discussing the details of the proposed method. This part has a fundamental weakness that the authors must give a more detailed explanation.
- The experiments can be designed in a more elaborate way to cover all important aspects of the proposed method. It is suggested that the authors provide more discussion on the results. Strengthen the discussion literature as the theoretical and practical implications are suggestive.
- The superiority of the proposed method should be addressed perfectly in comparison with similar established ones in the literature.
- The conclusion section should not be a summary of the work, as the abstract. This is a synthesis of the key points of the work which respond to the research question/s that should be posted in the introduction section. The impact of the work in the general context should be highlighted and Future work should be suggested.
- The authors cited only two papers from 2020-2022. Please discuss and cite more papers from 2020-2022.
Reviewer 4 Report
Very good methodological approach.
Minor English corrections required.
Round 2
Reviewer 3 Report
The authors revised the paper properly and I have no further comments. Therefore, I recommend the acceptance of the paper at this stage.